# Electronic Cigarettes and Asthma: What Do We Know So Far?

**DOI:** 10.3390/jpm11080723

**Published:** 2021-07-27

**Authors:** Serafeim-Chrysovalantis Kotoulas, Paraskevi Katsaounou, Renata Riha, Ioanna Grigoriou, Despoina Papakosta, Dionysios Spyratos, Konstantinos Porpodis, Kalliopi Domvri, Athanasia Pataka

**Affiliations:** 1Clinic of Respiratory Failure, General Hospital of Thessaloniki Georgios Papanikolaou, Aristotle University of Thessaloniki, Leoforos Papanikolaou, 57010 Thessaloniki, Greece; ioagrig@hotmail.gr (I.G.); patakath@yahoo.gr (A.P.); 21st ICU “Evangelismos Hospital”, School of Medicine, National and Kapodistrian University of Athens, Ypsilantou 45-47, 10676 Athens, Greece; paraskevikatsaounou@gmail.com; 3Sleep Research Unit, Department of Sleep Medicine, The University of Edinburgh, Royal Infirmary of Edinburgh, 51 Little France Crescent, Edinburgh EH16 4SA, UK; rlriha@hotmail.com; 4Department of Pulmonary Medicine, General Hospital of Thessaloniki “Georgios Papanikolaou”, Aristotle University of Thessaloniki, Leoforos Papanikolaou, 57010 Thessaloniki, Greece; depapako@gmail.com (D.P.); diospyrato@yahoo.gr (D.S.); kporpodis@yahoo.gr (K.P.); kellybio4@hotmail.com (K.D.)

**Keywords:** electronic cigarette, asthma, review, asthma pathogenesis, lung function, airway inflammation, asthma control, smoking cessation

## Abstract

Electronic cigarettes (EC) are a novel product, marketed as an alternative to tobacco cigarette. Its effects on human health have not been investigated widely yet, especially in specific populations such as patients with asthma. With this review, we use the existing literature in order to answer four crucial questions concerning: (1) ECs’ role in the pathogenesis of asthma; (2) ECs’ effects on lung function and airway inflammation in patients with asthma; (3) ECs’ effects on asthma clinical characteristics in asthmatics who use it regularly; and (4) ECs’ effectiveness as a smoking cessation tool in these patients. Evidence suggests that many EC compounds might contribute to the pathogenesis of asthma. Lung function seems to deteriorate by the use of EC in this population, while airway inflammation alters, with the aggravation of T-helper-type-2 (Th2) inflammation being the most prominent but not the exclusive effect. EC also seems to worsen asthma symptoms and the rate and severity of exacerbations in asthmatics who are current vapers, whilst evidence suggests that its effectiveness as a smoking cessation tool might be limited. Asthmatic patients should avoid using EC.

## 1. Introduction

Asthma is a chronic inflammatory airway disease characterized by exacerbations and remissions, affecting 1–18% of the population in different countries [1]. Airway inflammation in asthma is typically eosinophilic, but can also be neutrophilic, mixed, or non-granulocytic [1]. Different factors are involved in aggravating airways inflammation in asthmatic patients, with cigarette smoke being one of the main ones [1,2]. Smoking prevalence in patients with asthma approximates that of the general population [3]. Even in severe asthma, the percentages of asthmatic smokers could reach or even exceed these of the general population [4]. The percentage of severe asthmatics that were using e-cigarettes (ECs) in the same study was 2% and was higher than in the general population in many countries [4]. Asthmatic smokers suffer from more symptoms and exacerbations, develop a more rapid decline in pulmonary function and have a worse prognosis than non-smoking asthmatics [5,6,7]. In addition, they usually present with a higher proportion of neutrophils in induced sputum, reduced pH, and heterogenous inflammatory mediator profiles in exhaled breath condensate (EBC) [8,9,10,11].

Since 2003, the EC has become very popular as it was advertised as a tool for smoking cessation. Clinical experience has shown that smokers attempting to quit smoking use the EC as an alternative [12,13,14,15]. However, ECs’ safety has not been scientifically demonstrated, especially in asthmatic patients. In this review, we consolidate the current knowledge about the effects of ECs on asthma by addressing four critical questions: (A) Do EC compounds play a role in the pathogenesis of asthma? (B) What is the effect of EC use on lung function and on airway inflammation in patients with asthma? (C) What is the effect of EC on the clinical characteristics of asthma? (D) Is EC use an effective strategy for smoking cessation in patients with asthma?

## 2. Materials and Methods

We performed a PubMed search in “All Fields” for “electronic nicotine delivery system” OR “electronic cigarette” OR “e-cigarette” AND “asthma” to the 31 March 2021 with no start date. This search identified 256 unique results. Two of the co-authors performed the PubMed search and identified the results. They decided which studies to exclude from this review based on their abstracts. Items were removed if they were editorials, comments on articles, meeting abstracts, a duplicate, not in English, or irrelevant to the research theme. Subsequently, the results of the procedure were presented to the other co-authors. There were no disagreements between the co-authors about the studies which included and excluded. As a result, 154 items were removed and 102 were included in this review. Articles were categorized based on the questions of this review. For the first question, 28 studies were used (15 reviews, 11 experimental studies, and 2 systematic reviews). For the second question, 15 studies were used (5 interventional animal studies, 4 interventional human studies, 3 observational studies, 2 case studies, and 1 systematic review). For the third question, 60 studies were used (39 observational studies, 8 reviews, 6 case studies, 5 systematic reviews with 4 meta-analyses, and 2 opinion articles). For the fourth question, 41 studies were used (25 observational studies, 8 reviews, 4 systematic reviews with 3 meta-analyses, 3 case studies, and 1 opinion article). Some of the studies included, were used to answer more than one question. For any studies written by one or more co-authors of the present review, an independent author, who did not participate in the study, reviewed it for suitability of inclusion.

## 3. Results

### 3.1. Do EC Compounds Play a Role in the Pathogenesis of Asthma?

The constitution of EC aerosol is defined by temperature, and by the contents in the heated liquid as propylene glycol (PG), glycerin, flavoring agents, nicotine in variable concentrations and other non-nicotine substances [16]. Laboratory, observational, and clinical studies have revealed that EC aerosols contain numerous respiratory irritants and toxins and that may have a cytotoxic effect on lung tissue, analogous to that of the tobacco cigarette [17,18]. More than 80 compounds (including known toxins—e.g., formaldehyde, acetaldehyde, metallic nanoparticles, and acrolein) have been found in e-liquid and aerosols and as a result, ECs have been linked with an increase in symptoms in individuals with asthma [16]. Additionally, ECs were found to contain not only formaldehyde but also formaldehyde-forming hemiacetals and potentially toxic particulate matter that deposits on surfaces [19]. The novel-generation high-power electronic nicotine delivery systems (ENDS) which seem to be particularly user-adaptive, produce droplets with a diameter at 0.78 ± 0.03 μm [20,21]. Exposure of the airway epithelial cells to certain liquid flavorings reaches toxicity thresholds. The chocolate flavoring 2,5-dimethypyrazine activates the cystic fibrosis transmembrane conductance regulator (CFTR) ion channel [22]. Work-related inhalation of several usual food-safe flavoring substances has been related with occupational asthma and asthma symptoms deterioration [23]. More specifically, work-related inhalation exposures to the flavoring substance diacetyl was found to cause irreversible obstructive airway disease in healthy workers. The thermal decomposition of PG and vegetable glycerin (VG), the key elements of EC liquids, generates reactive carbonyls, including acetaldehyde, formaldehyde, and acrolein which have well-known lung toxicities [23]. PG vapor has been found to induce respiratory irritation and increase asthma risk, despite the fact that EC use improved home indoor air quality compared with secondhand tobacco smoke [24]. Long-term exposure to EC was found to change the human bronchial epithelial proteome promoting its damage [25]. Heavy EC smoking promotes inflammatory processes (activator of transcription and nuclear factor-κB signaling, Janus tyrosine kinase/signal transducer, and mitogen-activated protein kinase), in a similar way to tobacco smoke. Protracted exposure to some components of EC vapor results in respiratory complications as asthma [26].

Chronic EC exposure also seems to result in increased neutrophil elastase and matrix metalloprotease levels in the lung, abnormal activation of the lung epithelial cells, β-defensins and neutrophilic response (NETosis), activation of transient receptor potential ankyrin 1 (TRAP1), alternations in the normal respiratory microbiota, induced proteolysis and in general impaired respiratory innate immune system, all associated with allergies and asthma [27,28]. Respiratory innate immune cell function has also been found to be impaired by flavored EC liquids and more specifically, by cinnamaldehyde which suppresses phagocytosis by macrophages [29], and provisionally represses ciliary mobility of bronchial epithelial cells through dysregulation of mitochondrial function [30]. These dysregulations of the respiratory immunity by EC could impact asthma development, severity, and/or exacerbations [31].

A recently published review [32] cited that ECs induced oxidative stress toxicity in a range of cells (including lung cells), released pro-inflammatory cytokines, and impaired the ability to fight infection. This increased pro-inflammatory activity suggests that switching to EC would not necessarily resolve the lung inflammation in smokers, while flavor additives could additionally affect cellular function by inducing phagocytosis and cytokine production [32]. Moreover, hydroxyl radical (^−^OH), the most destructive of reactive oxygen species (ROS), was found in significant amounts in EC vapor due to, or depending on, increased power output, puff volume, coil temperature, and oxygen supply. VG produced greater ^−^OH levels than PG and flavored liquids compared to non-flavored ones. Subsequently, even though the dose of ^−^OH levels per puff associated with EC vaping was 10–1000 times lower than the reported dose generated by conventional cigarette, the daily average ^−^OH level could be comparable to that from cigarette smoking depending on vaping patterns. This means that EC users vaping VG-based flavored liquids at higher power output settings may be at increased risk for ^−^OH exposures and related health consequences, such as asthma [33]. Apart from oxidative stress agents, EC has been shown to promote disorders in pro-inflammatory and pro-fibrotic markers, DNA damage and dysfunction of DNA repair and antioxidant enzymes, which participate in lung disease pathogenesis, such as asthma [34]. Long non-coding RNAs (lncRNAs), which are involved in cell proliferation and differentiation, intracellular communication, and steroid metabolism in lung pathologies, including asthma, have also been found altered after exposure to EC vapor [35].

A recent study has shown that EC smokers were at higher risk for developing diseases—such as asthma, coronary artery disease, and lung cancer—than their non-smoking counterparts as downstream metabolites derived from nicotine were found in ECs [36]. Nicotine was found to be the key agent affecting in utero lung development of the fetus, contributing to the increased prevalence of childhood diseases such as asthma in smokers [37]. Nicotine induces cell-specific molecular changes in the lungs [38]. Prenatal exposure to nicotine provoked epigenetic reprogramming in the new-born child, anomalous respiratory growth, and multigenerational transmission of symptomatology similar to that of asthma [23]. In utero exposure to nicotine-containing EC in murine models increased the risk of asthma in offsprings by enhancing T-helper-type-2 (Th2)-mediated inflammation, changing mitochondrial homeostasis and impairing airway cell function [39]. Some e-liquid flavoring agents, such as aromatic aldehydes, may reduce Cytochrome P450 2A6 (CYP2A6) activity, an enzyme that metabolizes almost 80% of nicotine, and consequently prolong nicotine activity [40]. EC usage during pregnancy is likely to be as dangerous to fetal lung development as maternal smoking [37]. Tobacco use and exposure are risk factors with potentially durable impact on lung health in early life. EC sales should thus be restricted during pregnancy [41]. Significant concerns are raised about the dual use of ECs and conventional cigarettes during pregnancy, as well as the use of ECs only, during pregnancy by women who think ECs are safe or they are not able to quit [42]. However, since nicotine itself has an additive negative impact to fetus, the complete abstinence of pregnant women from all nicotine-contained products is absolutely necessary [43]. Moreover, a review on the epigenetic impacts of maternal tobacco and e-vapor exposure on offspring lungs, concluded that maternal EC use, not only induces epigenetic changes that could contribute to developing asthma, but these changes could possibly be passed down to further generations independent of exposure [44].

The possible mechanisms whereby EC compounds can affect asthma are shown in Figure 1.

### 3.2. What Is the Effect of EC Use on Lung Function and on Airway Inflammation in Patients with Asthma?

Studies on the effects of EC on the lung function of asthmatic patients are few and even fewer regarding airway inflammation (Table 1). Asthmatic patients exhibited a significant increase in respiratory system total impedance at 5 Hz (Z5), respiratory resistance at 5, 10, and 20 Hz (R5, R10, and R20), resonant frequency and reactance area measured by impulse oscillometry (IOS) after EC use, compared with healthy controls [45]. Mean airway resistance along with the slope of the phase III curve on the single breath nitrogen test increased immediately after short-term EC use in a group of asthmatic smokers, thereby demonstrating airway dysfunction, particularly in small airways [46]. Apart from airway resistance, asthmatic patients also exhibited impaired pulmonary function tests (PFTs) after vaping for five minutes, with the decrease in forced expiratory volume in 1 s to forced vital capacity ratio (FEV1/FVC) and peak expiratory flow (PEF) being more significant [47]. Furthermore, patients who recovered from electronic vapor acute lung injury (EVALI), a condition more commonly observed in asthmatic patients, exhibited chronic irreversible airflow obstruction, markedly abnormal ^129^Xe MRI ventilation heterogeneity, abnormal lung clearance index and oscillometry measures and decreased diffusing capacity of the lung for carbon monoxide (DLCO), all persistent after their discharge [48,49]. Studies in animals with allergen-induced airway disease demonstrated not only increased airway hyperresponsiveness after EC vapor inhalation, but also increase in mucus and airway wall thickening which are hallmark features of allergic asthma [34,39,50,51].

On the other hand, there is evidence from a small group of asthmatics (10 patients) that the use of ECs free of nicotine, filled with a mixture of PG and glycerol, did not significantly affect pulmonary function and symptoms [52]. In a study of only 15 asthmatic patients that switched from conventional cigarettes to electronic, PFTs remained stable over three visits [53]. Furthermore, two studies reported that lung function in the same 18 asthmatic patients who switched from tobacco cigarette to EC and were monitored for 30–36 months improved [54,55]. However, apart from the fact that the above two studies included a small number of the same patients, there are serious concerns about selection bias since those patients were reviewed retrospectively [54], and then prospectively [55].

The effects of EC use on inflammation have been studied in cell lines, animal models, and humans. In all three, EC use led to inflammation and oxidative stress [56]. However, specifically in asthmatic patients, the studies that evaluate the effects of ECs on airway inflammation are limited [32]. ECs free of nicotine were found to cause heterogenous effects depending on their flavor, while ECs containing nicotine suppressed airway inflammation but not airway remodeling in mice with allergic airway disease [50]. Eosinophilic inflammation is accompanied by an increased fraction of exhaled nitric oxide (FeNO) and correlates with other indices of inflammation in asthmatic patients [57]. FeNO is increased during asthma exacerbations, while it decreases with recovery or inhaled corticosteroids [58,59]. There is conflicting evidence on the effect of EC on the FeNO of asthmatics. There are studies where FeNO significantly decreased after an EC session [45], whereas the opposite result was exhibited in another study [47]. In the latter study, Th2 cytokines such as interleukins (IL) IL-4 and IL-13 in the EBC of asthmatics were found to be significantly increased after vaping for five minutes, reflecting increased eosinophilic inflammation, and supporting the finding of increased FeNO [47]. Apart from Th2 inflammatory mediators, an increase in IL-1β and tumor necrosis factor alpha (TNF-α) was observed. Both are proinflammatory cytokines that amplify and orchestrate the inflammatory response in asthma and determine its severity; IL-10, a cytokine derived from Th2 cells and 8-Isoprostane (ISO8) a biomarker of oxidative stress were also increased [47]. Additionally, in three experimental studies on mice with allergen-induced airway disease, EC inhalation increased infiltration of what by inflammatory cells, including eosinophils, into airways from blood, increased the number of all types of inflammatory cells in Bronchoalveolar lavage fluid (BALF), stimulated the production of Th2 cytokines such as IL-4, IL-5, and IL-13 and allergen-specific immunoglobulin E (IgE) and reduced the levels of transforming growth factor (TGF)-β1 and matrix metalloproteinase (MMP)-2 in lung tissue homogenate [39,51,60].

**Table 1 jpm-11-00723-t001:** Studies addressing research question 2: What is the effect of EC use on lung function and on airway inflammation in patients with asthma?

Reference	Study Type	Research Orientation	Main Findings	Main Limitations
Lappas 2017 [45]	Interventional human study	Lung function and airway inflammation	IOS parameters deteriorated acutely after a session of EC, with the changes being more prominent in asthmatic patients compared with healthy controls. FeNO decreased significantly in both groups and remained lower for a greater period in asthmatics compared with healthy controls.	Small number of participants (27 smokers with mild asthma and 27 healthy smokers).
Palamidas 2017 [46]	Interventional human study	Lung function	Airway resistances and small airway function deteriorated significantly in a group of asthmatics after an EC session.	Small number of participants with asthma (11).
Kotoulas 2020 [47]	Interventional human study	Lung function and airway inflammation	IOS parameters deteriorated acutely after a session of EC in both groups. PEF and FEV1/FVC deteriorated significantly after a session of EC in patients with asthma but not in healthy controls. FeNO and Th2 cytokines (IL-4 and IL-13) increased significantly in asthmatic patients after an EC session compared with controls. IL-1β, TNF-α, IL-10 and ISO8 increased significantly in asthmatic patients after an EC session compared with controls.	Small number of participants (25 smokers with moderate asthma and 25 healthy smokers).
Eddy 2020 [48]	Case report	Lung function	A patient who recovered from EVALI exhibited chronic irreversible airflow obstruction, markedly abnormal ^129^Xe MRI ventilation heterogeneity and abnormal lung clearance index and oscillometry measures 8 months after discharge.	Case report.
Reddy 2021 [49]	Case report	Lung function	Two patients who recovered from EVALI showed decreased diffusing capacity of the lung for carbon monoxide.	Case report.
Marczylo 2020 [34]	Review of animal studies	Lung function and airway inflammation	Increase in airway hyperresponsiveness, on exposure to electronic cigarettes, across mouse strains, sex, and ages.	Animal studies.
McAlinden 2020 [39]	Interventional animal study	Lung function and airway inflammation	Allergen-challenge in mice lead to significant increase in airway inflammation (mainly th2-dependent), development of airway hyperresponsiveness and increase in mucus and airway wall thickening.	Animal study.
Chapman 2019 [50]	Interventional animal study	Lung function and airway inflammation	Increased or stable peripheral airway hyperresponsiveness after exposion to EC aerosol depending on EC’s flavor. Suppressed airway inflammation after exposion to EC aerosol.	Animal study.
Lim 2014 [51]	Interventional animal study	Lung function and airway inflammation	Increased airway hyperresponsiveness after EC vapor inhalation. Increased infiltration of inflammatory cells, including eosinophils, into airways from blood. Stimulation of production of Th2 cytokines such as IL-4, IL-5 and IL-13 and allergen-specific IgE.	Animal study.
Boulay 2017 [52]	Interventional human study	Lung function	Pulmonary function was not significantly affected after the use of ECs free of nicotine, filled with a mixture of propylene glycol and glycerol.	Small number of participants with asthma (10).
Solinas 2020 [53]	Observational human study	Lung function	PFTs remained stable throughout three visits in a group of asthmatic patients that switched from conventional cigarette to electronic.	Small number of participants with asthma (10).
Polosa 2014 [54]	Observational human study	Lung function	Lung function improved in a group of asthmatic patients after switching from tobacco cigarette to EC.	Small number of participants with asthma (18). Possible selection bias.
Polosa 2016 [55]	Observational human study	Lung function	Lung function improved in a group of asthmatic patients after switching from tobacco cigarette to EC.	Small number of participants with asthma (18). Possible selection bias.
Bozier 2020 [32]	Systematic review	Lung function	The studies included in this review concluded that lung function was deteriorated acutely after EC use.	The study investigated the effects of EC in general and was not emphasized in a certain group like patients with asthma.
Taha 2020 [60]	Interventional animal study	Airway inflammation	EC aerosol significantly increased the number of all types of inflammatory cells in BALF and their airway recruitment, reduced the levels of TGF-β1 and MMP-2 in lung tissue homogenate and increased the level of IL-13 in airways.	Animal study.
Concluding paragraph: Most studies suggest that EC acutely deteriorates lung function in patients with asthma. Studies that concluded in no difference, or even in improvement, exhibited serious methodological errors and included a small number of participants. Airway inflammation was also found to be altered, mainly the Th2 inflammatory pathway, but not limited to that.				

EC = electronic cigarette, IOS = impulse oscillometry, FeNO = fraction of exhaled Nitric Oxide, PEF = peak expiratory flow, FEV1/FVC = forced expiratory volume in 1 s to forced vital capacity ratio, Th2 = T-helper-type-2, IL = Interleukin, TNF-α = tumor necrosis factor-α, ISO8 = 8-Isoprostane, EVALI = electronic vapor acute lung injury, 129Xe MRI = magnetic resonance imaging with xenon-129, IgE = Immunoglobulin E, PFTs = pulmonary function tests, BALF = Bronchoalveolar lavage fluid, TGF = transforming growth factor, MMP = matrix metalloproteinase.

### 3.3. What Is the Effect of EC on the Clinical Characteristics of Asthma?

Thirty-nine observational studies including 2,111,023 participants, six case studies, two opinion articles, eight reviews, and five systematic reviews with four meta-analyses investigated the effects of EC use in asthmatics (Table 2). Several investigators have concluded that EC could be associated with the development of pulmonary disorders, including asthma and might increase asthma severity and exacerbations [18,31,61,62,63,64]. Numerous cross-sectional studies with a large number of participants have described the significant association between EC use and even secondhand exposure and asthma diagnosis and severity [65,66,67,68,69,70,71,72,73,74], compared to the few studies which found no association [75,76], or even negative association between EC use and asthma [77,78]. A prospective cohort study also found that EC use was associated with an increased risk of developing respiratory disease, including asthma, independent of cigarette smoking [79]. A study from Korea demonstrated that adolescent EC users presented the highest adjusted odds ratio for severe asthma, which was reflected by the number of days absent from school due to asthma symptoms [80]. EC use was found to be positively correlated with asthma, or even more, to increase the probability of an adolescent being diagnosed with asthma and also enhanced the adverse effects of tobacco cigarettes in asthma [80,81,82]. A study from Sweden which comprised patients with obstructive lung diseases, mostly asthmatics, showed that all respiratory symptoms were most common among dual users (electronic plus tobacco cigarette), former smokers and nonsmokers who used ECs rather than tobacco cigarette smokers-alone [83]. Furthermore, two studies from France and Canada also found that asthma was more commonly associated with EC use [84,85]. A large epidemiological study from the USA including more than 400,000 participants showed that current EC use was associated with 39% higher odds of self-reported asthma, compared to never EC use and that there was a graded increased odds of having asthma with increased EC use intensity, from occasional to daily EC users [86]. Five more studies from the USA also concluded that EC is an independent risk factor for respiratory disease including asthma, after controlling for covariates [87,88,89,90,91]. EC had an additive effect for asthma beyond smoking [91]. Dual use, which is the most common usage pattern, is riskier than using either product alone [87]. Dual use, with even passive exposure to EC, was identified as significant predictor for asthma in two more cross-sectional studies [73,92] and one meta-analysis [93]. A recently published systematic review concluded that evidence up to now suggests that the side effects of ECs may be exaggerated in people with asthma [32]. Additionally, asthma symptoms were among the most frequently reported side effects associated with EC use, second to headaches [94]. Moreover, EC use was associated with lower general health scores, higher breathing difficulty scores and a greater proportion of reporting asthma [95]. A large epidemiological study in USA with a weighted sample size of 31,721,603 adults between 18 and 24 years (2,503,503 with former and 3,200,681 with current asthma) found that the prevalence of EC use was significantly higher among young adults with current or former asthma and that asthma combined with EC use was significantly associated with worse mental health [96], a finding similar to that of another study from Korea [72].

There are studies about EC use for harm reduction in asthmatic cigarette smokers [97]. A study pointed out that EC is far less problematic compared with combustible cigarettes and that exclusive EC users had substantially lower risk of exposure to tobacco smoke toxins compared to cigarette smokers. There is also emerging evidence that switching to regular EC use could produce significant respiratory health gains [98]. A study from Italy demonstrated that 90% of 382 asthmatic-vapers-only declared that vaping did not worsen their asthma symptoms and would recommend EC use. However, the majority of participants responded to a web survey and only a small fraction (55 volunteers, including 10 asthmatics), reported improvement in their asthma control after switching to EC but without PFTs improvement over three visits [53]. Similar findings regarding asthma control improvement were demonstrated by another worldwide online survey and two more studies from Italy but with no changes in the exacerbation rates [54,55,99]. However, all these studies exhibited considerable selection bias (Table 2).

A systematic review with meta-analysis found that former smokers who transitioned to EC showed approximately 40% lower odds of respiratory outcomes, including asthma, compared to current exclusive smokers [100]. However, that meta-analysis included only six studies (five cross-sectional and only three reported respiratory outcomes), whereas three more recently published systematic reviews with meta-analyses which included more studies with a lower percentage of cross-sectional studies found a significant association between EC use and dual use with asthma [93], both diagnosis and exacerbations [101], and after controlling for cigarette smoking and other covariates [102]. Most of the studies that were included in those meta-analyses are also included in the present review (Table 2).

As far as for asthma exacerbations, it was reported that EC use was associated with the frequency of asthma attacks in the past 12 months [103]. Even second-hand EC exposure was associated with higher odds of reporting an asthma attack in adolescents [104] and children [105]. Additionally, several cases of EVALI (some of them mimicking COVID-19) were recently reported in patients with a history of asthma [49,62,106,107,108,109]. Asthma is more common in patients with EVALI (approximately 30%; 43.6% in adolescents and 28.3% in adults) compared to the 8% to 10% of the general population [109,110]. Furthermore, a higher proportion of fatal cases of EVALI had a history of asthma compared to non-fatal cases (23% vs. 8%) [111]. Moreover, extremely severe status asthmaticus with hypercarbia requiring veno-venous extracorporeal membrane oxygenation (VV-ECMO) and slow recovery on extensive bronchodilator and steroid regimens were also reported in two adolescents with a history of recent and past EC use and asthma [112]. ECMO was also necessary in a patient with a history of asthma who developed hypersensitivity pneumonitis (HP) secondary to EC use [113].

**Table 2 jpm-11-00723-t002:** Studies addressing research question 3: What is the effect of EC on the clinical characteristics of asthma?

Reference	Study Type	Participants	Main Findings	Main Limitations
Traboulsi 2020 [18]	Review	Previously published studies	The long-term health effects of EC are unknown as it can cause cellular alterations analogous to traditional tobacco smoke. Outlines possible clinical disorders associated with vaping on pulmonary health including asthma.	The study investigated several aspects regarding EC and did not emphasize in a certain group like patients with asthma.
Hickman 2020 [31]	Review	Previously published studies	Because respiratory immunity is already dysregulated in asthma, further alteration of cellular function by EC could impact asthma development, severity, and/or exacerbations.	Conclusions based on previously published studies and not on research data by controlled human exposure studies.
St Claire 2020 [61]	Review	Previously published studies	ENDS are associated with increased risk of lung disorders including asthma.	The study was oriented more towards tobacco cigarette and not EC.
Casey 2020 [62]	Review	Previously published studies	Details the described lung diseases associated with vaping with a focus on EVALI, and the predicted long-term consequences of EC use, including increased asthma severity.	The study was oriented more towards EVALI and not asthma.
Galderisi 2020 [63]	Opinion article	Previously published studies	Flavored EC liquids and aerosols contain airway irritants and toxicants that, in turn, produced an increase in asthma prevalence and its exacerbations among adolescents.	Opinion article.
Hernandez 2021 [64]	Review	Previously published studies	Steep rise in EC use among individuals with underlying lung disease, such as asthma.	The study investigated several aspects regarding EC and did not emphasize in a certain group like patients with asthma.
Entwistle 2020 [65]	Cross-sectional survey	1277 adults with asthma	EC use was associated with increased odds of having more frequent asthma symptoms.	Case-crossover study: A cause-and-effect relationship could not be established. Recall bias as participants reported their asthma status and did not report details about asthma phenotypes or control medications. There was no information about the amount and frequency of EC use and the dual use with tobacco cigarette.
Clawson 2020 [66]	Cross-sectional web-based survey	178 college students with asthma	40% of the participants had a history of EC use.	Case-crossover study: A cause-and-effect relationship could not be established.
Xie 2020 [67]	School based, cross-sectional, national representative study	12,747 high school students from the 2017 Youth Risk Behavior Survey	Overall, self-reported asthma prevalence estimates were significantly higher in current ENDs users compared with their ever- and never-used counterparts among US youth.	Case-crossover study: A cause-and-effect relationship could not be established.
Han 2020 [68]	Cross-sectional web-based survey	490,171 subjects (44,479 adolescents with physician-diagnosed asthma)	EC smoking behavior was significantly more frequent in adolescents with asthma than in those without asthma.	Case-crossover study: A cause-and-effect relationship could not be established. The self-reported nature introduces the possibility of misclassification in the dataset.
Alnajem 2020 [69]	A school-based cross-sectional study	1565 high school students (aged 16–19 years) in Kuwait	Compared to never EC users and never cigarette smokers, current EC users with no history of cigarette smoking had increased prevalence of current wheeze and current asthma. The frequency of exposure to household secondhand aerosols from EC was associated with current uncontrolled asthma symptoms.	Self-reporting of asthma symptoms may introduce information bias. Case-crossover study: A cause-and-effect relationship could not be established.
Parekh 2020 [70]	Cross-sectional database survey	131,965 women of childbearing age	Compared with nonsmokers, current EC users without a history of combustible cigarette smoking were associated with 74% higher odds of having asthma.	Case-crossover study: A cause-and-effect relationship could not be established.
Han 2020 [71]	Cross-sectional database survey	21,532 participants	In U.S. adolescents, use of an electronic vapor product was associated with lifetime asthma.	Case-crossover study: A cause-and-effect relationship could not be established.
Kim 2020 [72]	Cross-sectional web-based survey	195,847 adolescents (17,403 with asthma)	The rate of experience of EC use was higher among asthmatic respondents than non-asthmatic respondents. Asthmatic respondents with experience of EC use had a much higher proportion of negative mental health states including depression and suicidality than subjects without EC experience.	Case-crossover study: A cause-and-effect relationship could not be established. Did not separate currently active asthma from previous (but treated or inactive) asthma, current symptoms and severity, treatment and adherence. The quantity and duration of conventional cigarette or EC consumption were not assessed. Because of the self-reported nature of the survey, recall bias could not be eliminated.
Ebrahimi Kalan 2021 [73]	Cross-sectional survey	34,183 adolescents who were never-tobacco product users	Adolescents who reported currently having asthma were more likely to report living with someone who smokes cigarettes, hookah, and poly tobacco.	Case-crossover study: A cause-and-effect relationship could not be established.
Alqahtani 2021 [74]	Cross-sectional survey	Data from the 2018 Behavioral Risk Factor Surveillance System	The prevalence of lifetime EC use was higher among adults with chronic lung disease, including asthma, than among those without.	Case-crossover study: A cause-and-effect relationship could not be established.
Tran 2020 [75]	Cross-sectional database survey	186,036 adults who responded question about EC use (23,071 with asthma)	Adults with asthma had similar odds of every day EC use, but higher odds of EC use on some days compared to adults without asthma.	The self-reported nature introduces the possibility of misclassification in the dataset. Case-crossover study: A cause-and-effect relationship could not be established.
Walker 2021 [76]	Cross-sectional telephone survey	2387 participants 18–30 years from Kentucky (253 with current asthma)	ENDS use did not significantly increase the odds of asthma. Population attributable fraction of asthma due to ENDS was 0.4%.	Case-crossover study: A cause-and-effect relationship could not be established. Because of the self-reported nature of the survey, recall bias could not be eliminated.
Gibson-Young 2020 [77]	Cross-sectional web-based survey	2298 undergraduate college students (446 with asthma)	Asthma was a significant predictor in reporting lower perceived health status than students without asthma and perceived health status was a significant predictor of reporting fewer ever use of ENDS.	Case-crossover study: A cause-and-effect relationship could not be established. Limitation due to the statistical methodology that was used.
Alanazi 2021 [78]	Cross-sectional survey	283 youth and young adults from Alabama (151 with asthma)	Susceptibility to EC use and current use of EC were both lower among youth and young adults with asthma.	Case-crossover study: A cause-and-effect relationship could not be established.
Xie 2020 [79]	Prospective cohort study used data from a nationally representative cohort of US adults (PATH)	21,618 respondents aged 18 years and older at baseline	EC use was associated with an increased risk of developing respiratory disease, including asthma, independent of cigarette smoking.	Self-reported measures of EC and other tobacco product use and on the diagnosis of the respiratory disease, may be subject to recall bias. The data were observational in nature, and the follow-up period was relatively short; thus, the study could not establish causality.
Cho 2016 [80]	Cross-sectional web-based survey	35,904 high school students with asthma	Current EC users had a higher probability of being diagnosed with asthma compared to never EC users. Adolescent EC users presented the highest adjusted odds ratio for severe asthma, which was reflected by the number of days absent from school due to asthma symptoms.	Case-crossover study: A cause-and-effect relationship could not be established. Recall and selection bias due to being based in an on-line survey.
Chung 2020 [81]	Cross-sectional web-based survey	60,040 adolescents (5158 patients with ever diagnosis of asthma and 1532 with current asthma)	EC increased the probability of an adolescent to be diagnosed with asthma. EC implicated the enhancement of the adverse effects of tobacco cigarette in asthma.	Case-crossover study: A cause-and-effect relationship could not be established. Recall and selection bias due to being based in an on-line survey.
Kim 2017 [82]	Cross-sectional web-based survey	216,056 adolescents (4890 diagnosed with asthma the last 12 months)	EC showed positive relation with asthma.	Case-crossover study: A cause-and-effect relationship could not be established. Recall and selection bias due to being based in an on-line survey.
Hedman 2018 [83]	Cross-sectional population-based study	30,272 participants mainly patients with asthma	All respiratory symptoms were most common among dual users (electronic plus tobacco cigarette), former smokers and nonsmokers who used ECs rather than tobacco cigarette smokers-alone.	Case-crossover study: A cause-and-effect relationship could not be established. Adjusted analyses among EC users between former smokers and nonsmokers were not possible because of a relatively low prevalence of EC use in the total sample population. The low response rates may have caused selection bias and lack of representativeness.
Aljandaleh 2020 [84]	Community based cohort study	368 adults (39 patients with asthma)	Asthma was more commonly associated with EC use.	Sample was not representative of the general population of young adults. Due to small numbers of EC users, there might have been a lack of statistical power to study some rare phenomena. EC use data were self-reported, which could have generated recall bias.
Larsen 2016 [85]	Population based survey	6159 high school students (21.3% with asthma)	Adolescents with asthma had higher odds of smoking ECs.	Case-crossover study: A cause-and-effect relationship could not be established. The number of responders with asthma was low and may have caused a selection bias. Asthma was self-reported and might have over or under-represent actual prevalence of asthma.
Osei 2017 [86]	Cross-sectional telephone survey	402,822 participants (34,074 with asthma)	Current EC use was associated with 39% higher odds of self-reported asthma compared to never EC use. There was a graded increased odds of having asthma with the increase of EC use intensity from occasional to daily EC users.	The exposures and outcomes were self-reported. There was no data on EC use initiation, duration, intensity (puffs/day) and flavorings used. Case-crossover study: A cause-and-effect relationship could not be established. The possibility of a self-selection based on a pre-existing condition could not be discounted.
Bhatta 2020 [87]	Population-based, longitudinal study	32,320 participants (5466 with respiratory disease)	EC was an independent risk factor for respiratory disease including asthma, after controlling for various covariates. Dual use, which was the most common use pattern, was riskier than using either product alone.	Several respiratory conditions were combined to obtain enough events to achieve adequate power. Recall bias because the use of ECs, conventional cigarettes and other combustible tobacco products was self-reported as were clinical conditions.
Wills 2019 [88]	Cross-sectional random-dial telephone survey	8087 participants (17% ever had asthma)	EC was an independent risk factor for respiratory disease including asthma, after controlling for various covariates.	Case-crossover study: A cause-and-effect relationship could not be established. The study lacked a detailed measure of smoking history, and the survey did not include items on marijuana. The respiratory variables were based on self-report. Sample was not representative of the general population.
Schweitzer 2017 [89]	School-based cross-sectional data	6089 students (34% ever had asthma, 22% currently had asthma)	EC was an independent risk factor for asthma, after controlling for various covariates.	The data on asthma were based on self-reports. There was missing data for the asthma measures. Additional indices of socioeconomic status and more extensive data on residential context and family hardship should have been measured. Case-crossover study: A cause-and-effect relationship could not be established.
Fedele 2016 [90]	Cross-sectional school-based paper-and-pencil questionnaire	32,414 high school students (3318 with asthma)	EC was an independent risk factor for asthma, after controlling for various covariates. Asthmatics were more likely to be current EC users compared to non-asthmatics.	Sample was not representative of the general population of young adults. Data were collected via adolescent self-report. The questionnaire did not include questions regarding the frequency of hookah or EC usage. Case-crossover study: A cause-and-effect relationship could not be established.
Wills 2020 [91]	School based, cross-sectional, national representative study	14,765 high school students from the 2017 Youth Risk Behavior Survey	EC is an independent risk factor for asthma, after controlling for various covariates. EC had an additive effect for asthma beyond smoking.	Case-crossover study: A cause-and-effect relationship could not be established. The survey did not include an item on currently having asthma and may represent an underestimate of effects. The survey did not ask about the type of EC device used and did not collect data on second-hand smoke exposure and household conditions.
Leavens 2020 [92]	Cross-sectional statewide survey	7775 adults who have experienced homelessness in Minnesota	Dual users had significantly higher rates of asthma than both those using combustible cigarettes and those using neither combustible nor EC.	Data were limited by being self-reported. Case-crossover study: A cause-and-effect relationship could not be established.
Xian 2021 [93]	Systematic review and meta-analysis	11 previously published cross-sectional studies including 1,143,118 participants	Significant association of both current and former EC use with asthma. Dual use had higher association odds with asthma than that of tobacco cigarette alone.	The utility of cross-sectional studies for causal inference is limited.
Bozier 2020 [32]	Systematic review	11 previously published studies from PubMed	The side effects of ECs may be exaggerated in people with asthma.	The study investigated the effects of EC in general and was not emphasized in a certain group like patients with asthma.
Hua 2020 [94]	On-line forum data extraction and analysis	41,216 posts with health effects produced by ECs (916 about asthma)	Asthma was among the most frequently reported disorder associated with EC use, second after headaches.	Data might have underestimated positive health effects, which EC users are less likely to post on online forums. The factors causing the symptoms and disorders reported by EC users could be complex. Demographic data on the study population were not extractable. It was not known if any individuals were dual users or if they had preexisting health conditions that may have affected their response to EC.
Wang 2018 [95]	Cross-sectional study using data from a longitudinal cohort	39,747 participants (3701 patients with asthma)	EC use was associated with lower general health scores, higher breathing difficulty scores, and greater proportions of reporting to have asthma. ECs alone may have contributed to increased respiratory health risks.	Case-crossover study: A cause-and-effect relationship could not be established. The timing of EC initiation was not available. Sample was not representative of the general population, and this could have caused selection bias. A validated measure of EC dose was not available. Self-reported outcomes might have resulted in recall bias. The study did not include assessment of baseline nicotine dependence levels.
Alanazi 2020 [96]	Cross-sectional database survey	Young adults between 18 and 24 years from the 2018 Behavioral Risk Factor Surveillance System of the US CDC (Weighted N = 31,721,603 adults, 2,503,503 with former and 3,200,681 with current asthma)	The prevalence of EC use was significantly higher among young adults with current or former asthma than those without asthma. The higher prevalence of EC use among those with current or former asthma was statistically accounted for by a greater number of bad mental health days in the past 30 days.	The data analyzed in this study were generated from a cross-sectional population survey, limiting the ability to infer causality. The responses were self-reported, which increases the risk of social desirability and other biases. Asthma diagnosis was based on a subjective measure, not a clinical diagnosis that also indicates the degree of severity, acuity, and treatment status.
Lee-Sarwar 2017 [97]	Review	Previously published studies	EC use may have a role in harm reduction for conventional cigarette smokers with asthma. Short-term and long-term effects of EC must be clarified.	This review investigated strategies to alter the natural history of childhood asthma in general and not its relationship with EC. The authors concluded that factors other than EC are more important in altering the natural history of childhood asthma, while the role of EC is undefined.
Polosa 2016 [98]	Review	Peer-reviewed articles from the PubMed about ECs	Vapor toxicology is far less problematic compared with combustible cigarettes. Exclusive EC users have substantial lower risk of exposure to tobacco smoke toxicants and carcinogens compared with cigarette smokers.	Under the heading: “What about Potential Harm?” the authors discussed about ECs’ benefits. Potential harms of ECs were hardly mentioned or were not mentioned at all. Many studies about ECs’ harms were not included in the study.
Solinas 2020 [53]	On-line based survey	2842 (646 patients with asthma)	Switching from tobacco cigarette to electronic did not worsen their asthma symptoms.	Selection bias due to being based in an on-line survey.
Polosa 2014 [54]	Retrospective observational study	18 asthmatics who switched from tobacco cigarette to EC	Many aspects of asthma control improved after asthma patients switched from tobacco cigarette to EC, but exacerbation rate did not.	Few patients with asthma for an observational study. Possible selection bias.
Polosa 2016 [55]	Prospective observational study	18 asthmatics who switched from tobacco cigarette to EC	Many aspects of asthma control improved after asthma patients switched from tobacco cigarette to EC, but exacerbation rate did not.	Few patients with asthma for an observational study. Possible selection bias.
Farsalinos 2014 [99]	On-line based survey	19,414 (1308 patients with asthma)	Significant benefits in physical status and improvements in pre-existing disease conditions including asthma.	Selection bias due to being based in an on-line survey.
Goniewicz 2020 [100]	Systematic review and meta-analysis	Six population-based studies (5 cross-sectional and 1 longitudinal–3 reported respiratory outcomes) with sample sizes ranging from 19,475 to 161,529 respondents	Former smokers who transitioned to EC showed ~ 40% lower odds of respiratory outcomes, including asthma, compared to current exclusive smokers.	A small number of mainly cross-sectional studies were included in this systematic review and only 3 reported respiratory outcomes. The utility of cross-sectional studies for causal inference is limited.
Gugala 2021 [101]	Systematic review and meta-analysis	45 studies (14 randomized experimental, 7 non-randomized experimental, 6 cohort, and 18 cross-sectional) including 1,465,292 participants	Association between EC use and pulmonary symptoms, asthma diagnosis and exacerbations.	Some of the studies which included in this meta-analysis were cross-sectional and their utility for causal inference was limited.
Wills 2021 [102]	Literature review and meta-analysis	15 previously published studies both cross-sectional and longitudinal	Epidemiological studies, both cross-sectional and longitudinal, show a significant association of EC use with asthma, controlling for cigarette smoking and other covariates.	Some of the studies which included in this meta-analysis were cross-sectional and their utility for causal inference was limited.
Choi 2016 [103]	School-based cross-sectional survey	36,085 participants (11.3% currently had asthma)	Asthmatics were more likely to be current EC users compared to non-asthmatics. EC use the previous 30 days before the study was associated with having an asthma attack in the past 12 months.	Sample was not representative of the general population of young adults. Case-crossover study: A cause-and-effect relationship could not be established. The prevalence did not represent all ENDS. The survey was conducted during the time when the EC market was emerging. The study was unable to control for socioeconomic status.
Bayly 2019 [104]	School-based cross-sectional survey	11,830 youths between 11 and 17 years with a self-reported diagnosis of asthma	Secondhand EC exposure was associated with higher odds of reporting an asthma attack the past 12 months, adjusting for covariates.	Case-crossover study: A cause-and-effect relationship could not be established. Sample was not representative of the general population of young adults. Data were limited by being self-reported.
Berlinski 2020 [105]	Review	Previously published studies	Increased risk of an asthma exacerbation was reported in children exposed to ENDS.	This review did not investigate the relationship between asthma and EC.
Reddy 2021 [49]	Case series	6 patients (3 with asthma)	6 patients with a median age of 17 years presented with EVALI and half of them had a preexisting diagnosis of asthma.	Case series.
Aberegg 2020 [106]	Case report	1 patient with asthma	A 23 year-old man with history of childhood asthma suffered from EVALI.	Case report.
Rodriguez 2020 [107]	Case report	1 patient with a history of childhood asthma	A 23-year-old male, with a history of childhood asthma developed EVALI mimicking COVID-19 disease.	Case report.
Chawla 2020 [108]	Case report	1 patient with a history of mild intermittent asthma	A 15-year-old male with a history of mild intermittent asthma presented with EVALI.	Case report.
Clapp 2020 [109]	Opinion article	Previously published studies	Underlying asthma was reported in 30% of EC associated hospitalizations, which is much higher than the 8% to 10% of asthmatic patients seen in the general population.	Opinion article.
Adkins 2020 [110]	Cross-sectional data-collection survey	2155 patients with EVALI (360 hospitalized or deceased adolescents, 859 young adults and 936 adults)	A history of asthma was more likely to be reported among adolescents (43.6%) than adults (28.3%), both much higher than the population average.	Data collection methods varied, which may have resulted in reporting inconsistencies that could not be accounted. Information was obtained from medical records and patient or proxy interviews, which may be incomplete or subject to social desirability or recall bias. Comparisons with national prevalence data may be inexact. This study only included hospitalized or deceased EVALI cases; it is possible that many more adolescents may have been affected but not severely enough to require hospitalization.
Werner 2020 [111]	Medical record based epidemiological study	2558 patients hospitalized for EVALI (115 had a history of asthma)	A higher proportion of those with fatal cases of EVALI had a history of asthma compared to those with non-fatal EVALI cases (23% vs. 8%).	It could not be established a causative relationship between the history of asthma and the risk of death from EVALI. The number of asthmatic patients was relatively small compared to the total number of patients.
Bradford 2019 [112]	Case series	2 patients with asthma	Two patients with history of recent and past EC use and asthma experienced an extremely severe status asthmaticus with hypercarbia requiring VV-ECMO and slow recovery on extensive bronchodilator and steroid regimens.	Case series.
Nair 2020 [113]	Case report	1 patient with a putative diagnosis of asthma	Hypersensitivity pneumonitis in a young person secondary to vaping who required extracorporeal membrane oxygenation.	Case report.
Concluding paragraph: Most studies suggest that EC acutely deteriorates lung function in patients with asthma. Studies that concluded in no difference, or even in improvement, exhibited serious methodological errors and included a small number of participants. Airway inflammation was also found to be altered, mainly the Th2 inflammatory pathway, but not limited to that.				

EC = electronic cigarette, ENDS = electronic nicotine delivery system, EVALI = electronic vapor acute lung injury, PATH = Population Assessment of Tobacco and Health, CDC = center for disease control, COVID-19 = coronavirus disease 2019, VV-ECMO = veno-venous extracorporeal membrane oxygenation.

### 3.4. Is EC Use an Effective Strategy for Smoking Cessation in Patients with Asthma?

Twenty-five observational studies involving 966,376 participants, three case studies, one opinion article, eight reviews, and four systematic reviews with three meta-analyses were identified for this query (Table 3). Several researchers have pointed out that EC might lead previously “nicotine naïve youths” to tobacco cigarette use [18,62,63,64,114]. Tobacco cigarette use and asthma were found to be associated with lifetime EC use and young adults who used ECs persisted with smoking tobacco cigarettes [84]. Adolescents with asthma were more likely to be dependent on nicotine compared to their non-asthmatic counterparts [115,116]; they started smoking earlier because of curiosity about cigarettes and continued because it improved their stress and anxiety [116]. There is strong evidence that bad mental health is associated with EC in asthmatics as well as with tobacco cigarette and illicit substance use [92,96,110]. This population was at increased risk of dual use of conventional cigarettes and EC and those who used ECs presented higher tobacco cigarette smoking susceptibility [85,90,103,115]. Asthmatics that used ECs had significantly greater odds of exhibiting problematic smoking behaviors characterized by dual use of multiple tobacco products including waterpipes and marijuana [117]. Furthermore, dual users reported consuming a significantly greater number of cigarettes per day compared to cigarette only users and dual use was not associated with reduced exposure to cigarettes [95]. In addition to that, dual use showed the strongest association with lifetime asthma compared to other forms of EC use [71,92]. Moreover, in adults with asthma, current smokers were more likely to have tried ECs than former/never smokers [83,118], and smoking status was the most consistent predictor of EC use among all age groups [118]. ECs did not adequately serve as a smoking cessation tool in the asthmatic population [83], and EC use was significantly more common in adult smokers with one or more comorbidities, including asthma, versus those without comorbidities [119]. This was not the case in an on-line based survey from Japan, which however, included a small percentage of asthmatics [120].

It has been pointed out that ECs could serve as a gateway to tobacco cigarette smoking or illicit drugs (the “gateway effect”) [19,63]. One cohort study and two large cross-sectional studies—including more than 670,000 participants, with many asthmatics among them—have demonstrated the association between EC and tobacco smoking, snus, alcohol, and other substance use [68,75,114]. EC devices can contain cannabis-based products including tetrahydrocannabinol (THC), the psychoactive component of cannabis (marijuana) [18]. Several asthmatics have suffered from EVALI after the use of such devices [49,107,108]. Two more cross-sectional studies have demonstrated the association between EC and cannabis [66,110], whereas another cross-sectional study has shown the opposite [78]. A recently published review and meta-analysis, despite the high heterogeneity among studies (I^2^ = 88%), showed that non-smoker EC users had 4.59 increased odds (95% CI: 3.60 to 5.85) to become tobacco cigarette smokers at a later stage in their lives [121]. EC experimentation among never-smokers, especially children and adolescents lead to nicotine addiction and increased the possibility of becoming a regular smoker. This is a major public health issue taking into account the fact that the prevalence of ever use of ECs has been found to be higher in patients with asthma [16]. Smokers who quit smoking using ECs continue to vape for long periods after smoking cessation with all that this condition entails for lung health [32].

On the other hand, EC has been advertised as less harmful than tobacco cigarette [54,55,77,98,99]. Polosa et al., suggest that based on the existing evidence, EC should be used as a smoking cessation tool when counseling patients with asthma [98]. A study based on an internet survey, found an impressive reduction in smoking percentage and in the number of tobacco cigarettes (by approximately 80% in both) [99]. In two more studies from Italy, 18 asthmatic smokers that managed to switch from tobacco cigarettes to EC reported substantial health gains; however, two of them (11.1%) relapsed to tobacco cigarette [54,55]. Two recent systematic reviews with meta-analyses exhibited that former smokers who changed to EC showed better respiratory outcomes, including asthma, compared to current exclusive smokers [100,101]; however, dual use was associated with the worst outcomes as far as asthma [93,101]. This is particularly concerning since dual use seems to be the most common form of EC use [61]. Finally, three reviews that investigated the effectiveness of EC as a tool for smoking cessation in different groups of patients stated that the evidence on the use of EC as potential cessation tool is inconclusive, as few studies have been able to demonstrate an impact of EC use on harm reduction related to combustible cigarettes, which however does not address nicotine addiction, and that additional research was needed for patients with pulmonary diseases, including asthma [61,64,122]. Current evidence is inconclusive to recommend ECs as effective or safe smoking-cessation tools [123,124]. Importantly, ECs’ safety has not been proven and as they are increasingly used by young people or people not previously addicted to tobacco cigarette, they may prove to be a significant future health burden as far as nicotine addiction [123,124].

**Table 3 jpm-11-00723-t003:** Studies addressing research question 4: Is EC use an effective strategy for smoking cessation in patients with asthma?

Reference	Study Type	Participants	Main Findings	Main Limitations
Traboulsi 2020 [18]	Review	Previously published studies	EC use has rapidly increased among current and former smokers as well as youth who have never smoked. EC devices can contain cannabis-based products including tetrahydrocannabinol, the psychoactive component of cannabis (marijuana).	The study investigated several aspects regarding EC and did not emphasize in a certain group like patients with asthma.
Casey 2020 [62]	Review	Previously published studies	EC use has rapidly expanded both in adult smokers and previously nicotine naïve youths.	The study was oriented more towards EVALI and not asthma.
Galderisi 2020 [63]	Opinion article	Previously published studies	EC turned into a paradoxical preferential gateway to tobacco and nicotine initiation for adolescents naïve to tobacco. A marketing strategy on media and social network resulted in an unprecedented trend up in tobacco consumption among adolescents and gave rise to a new generation of nicotine-addicted teenagers.	Opinion article.
Hernandez 2021 [64]	Review	Previously published studies	Very few studies have been able to demonstrate an impact of EC use on harm reduction related to combustible cigarettes. ECs appear to serve as switching products that may help individuals reduce or quit cigarette use, but do not address nicotine addiction. Among the harms of EC use are included the nicotine dependence and the promotion of initiation of cigarette use amongst “never smokers”. There is a steep rise in EC use among teenagers and young adults who have never smoked.	The study investigated several aspects regarding EC and did not emphasize in a certain group like patients with asthma.
Hedman 2020 [114]	Cohort study about asthma and allergic diseases among school children	2185 participants recruited at age 7–8 years, and participated in questionnaire surveys at age 14–15 and 19 years	Among those who were daily tobacco smokers at age 14–15 years, 60.9% had tried EC at age 19 years compared with 19.1% of never-smokers and 34.0% of occasional smokers. EC use was associated with personal and parental tobacco use and use of snus. Almost one-third of those who had tried EC at age 19 years had never been tobacco smokers.	Tobacco and EC use was based on self-reports and not verified by objective measures such as level of cotinine. Did not include questions about personality traits related to tobacco or nicotine product initiation, sensation-seeking behavior, alcohol intake or other risk-taking behavior in the questionnaire.
Aljandaleh 2020 [84]	Community based cohort study	368 adults (39 patients with asthma)	Tobacco cigarette use and asthma were, among other factors, associated with lifetime EC use. Young adults who use ECs tend to persist in smoking tobacco cigarettes.	Sample was not representative of the general population of young adults. Due to small numbers of EC users, there might have been a lack of statistical power to study some rare phenomena. EC use data were self-reported, which could have generated recall bias.
Turner 2018 [115]	Longitudinal survey in a nationally representative cohort	1859 youth, with 19% (n = 353) reporting an asthma diagnosis	Asthmatic adolescents were significantly more likely to become addicted to EC compared to their non-asthmatic counterparts.	The cross-sectional design of the study limited the ability to determine the temporal sequences between asthma and EC. Did not examine EC use in combination with other tobacco products. Did not examine associations with asthma-related variables because of the small number of participants with asthma.
Vázquez-Nava 2017 [116]	Cross-sectional study	3383 adolescents (430 with asthma)	Adolescents with asthma are more likely to be dependent by nicotine compared to their healthy counterparts. They start smoking earlier because of curiosity about cigarettes. They continue smoking because this habit improves their sense of stress and anxiety.	Due to it being a case-crossover study, it could not establish a causative relationship between EC and asthma exacerbations. The smoking habits, the degree of nicotine addiction and the diagnosis of asthma were based on self-report questionnaires by the adolescent participants.
Leavens 2020 [92]	Cross-sectional statewide survey	7775 adults who have experienced homelessness in Minnesota	Among the strongest bivariate correlates of past 30-day EC use were substance use, mental health diagnosis and combustible cigarette smoking. Dual users had significantly higher rates of asthma.	Data were limited by being self-reported. Case-crossover study: A cause-and-effect relationship could not be established.
Alanazi 2020 [96]	Cross-sectional database survey	Young adults between 18 and 24 years from the 2018 Behavioral Risk Factor Surveillance System of the US CDC (Weighted N = 31,721,603 adults, 2,503,503 with former and 3,200,681 with current asthma)	The higher prevalence of EC use among those with current or former asthma was statistically accounted for by a greater number of bad mental health days in the past 30 days.	The data analyzed in this study were generated from a cross-sectional population survey, limiting the ability to infer causality. The responses were self-reported, which increases the risk of social desirability and other biases. Asthma diagnosis was based on a subjective measure, not a clinical diagnosis that also indicates the degree of severity, acuity and treatment status. The data could not specify the type of mental health problems since the measure was broad to stress, depression, and problems with emotions.
Adkins 2020 [110]	Cross-sectional data-collection survey	2155 patients with EVALI (360 hospitalized or deceased adolescents, 859 young adults and 936 adults)	Adolescents diagnosed as having EVALI reported using any nicotine-containing (62.4%), any tetrahydrocannabinol (THC)-containing (81.7%), and both (50.8%) types of EC or vaping products. Mental, emotional, or behavioral disorders were commonly reported.	Data collection methods varied, which may have resulted in reporting inconsistencies that could not be accounted. Information was obtained from medical records and patient or proxy interviews, which may be incomplete or subject to social desirability or recall bias. Comparisons with national prevalence data may be inexact. This study only included hospitalized or deceased EVALI cases; it is possible that many more adolescents may have been affected but not severely enough to require hospitalization.
Larsen 2016 [85]	Population based survey	6159 high school students (21.3% with asthma)	Adolescents with asthma were in increased risk for current use of tobacco and EC.	The cross-sectional design of the study limited the ability to determine the temporal sequences between tobacco cigarette and EC. The number of responders with asthma was low and may have caused a selection bias. Asthma was self-reported and might have over or under-represent actual prevalence of asthma.
Fedele 2016 [90]	Cross-sectional school-based paper-and-pencil questionnaire	32,414 high school students (3318 with asthma)	Adolescents with asthma were in increased risk for current use of tobacco and EC.	Sample was not representative of the general population of young adults. Data were collected via adolescent self-report. The questionnaire did not include questions regarding the frequency of hookah or EC usage. The cross-sectional design of the study limited the ability to determine the temporal sequences between tobacco cigarette and EC.
Choi 2016 [103]	School-based cross-sectional survey	36,085 participants (11.3% currently had asthma)	Adolescents with asthma were in increased risk for current use of tobacco and EC. Asthmatics who used ECs presented higher tobacco cigarette smoking susceptibility compared to non-asthmatics.	The cross-sectional design of the study limited the ability to determine the temporal sequences between tobacco cigarette and EC. Sample was not representative of the general population of young adults. The prevalence did not represent all ENDS. The survey was conducted during the time when the EC market was emerging. The study was unable to control for socioeconomic status.
Martinasek 2019 [117]	Cross-sectional online survey	898 college students (19.7% previously diagnosed with asthma)	Asthmatics who used ECs had significantly greater odds of exhibiting problematic smoking behaviors characterized by dual use of multiple tobacco products including tobacco cigarette hookah, and marijuana.	The cross-sectional design of the study limited the ability to determine the temporal sequences between tobacco cigarette and EC. Sample was not representative of the general population of young adults. The sample of those with asthma was small so some statistical tests may have been under powered to detect small differences. Data were collected via self-report.
Wang 2018 [95]	Cross-sectional study using data from a longitudinal cohort	39,747 participants (3701 patients with asthma)	Dual users reported significantly greater number of cigarettes per day compared to cigarette only users thus dual use was not associated with reduced exposure to cigarettes, compared to cigarette only users.	The cross-sectional design of the study limited the ability to determine the temporal sequences between tobacco cigarette and EC. The timing of EC initiation was not available. Sample was not representative of the general population, and this could have caused selection bias. A validated measure of EC dose was not available. Self-reported outcomes might have resulted in recall bias. The study did not include assessment of baseline nicotine dependence levels.
Han 2020 [71]	Cross-sectional database survey	21,532 participants	In U.S. adolescents, use of an electronic vapor product was associated with lifetime asthma, and this association was stronger when an electronic vapor product was used together with marijuana, particularly in combination with cigarette smoking.	Case-crossover study: A cause-and-effect relationship could not be established.
Hedman 2018 [83]	Cross-sectional population-based study	30,272 participants mainly patients with asthma	EC use was significantly more common among current smokers compared to former smokers and nonsmokers. ECs did not adequately serve as a smoking cessation tool in asthmatic population.	The cross-sectional design of the study limited the ability to determine the temporal sequences between tobacco cigarette and EC. Adjusted analyses among EC users between former smokers and nonsmokers were not possible because of a relatively low prevalence of EC use in the total sample population. The low response rates may have caused selection bias and lack of representativeness.
Deshpande 2020 [118]	Retrospective, cross-sectional study	10,578 adults with current asthma	Current asthmatic smokers were more likely to have tried ECs than former/never smokers. Smoking status was the most consistent predictor of EC use among all age groups.	The independent variables of the study were limited to the data that were available in the dataset. The study was unable to control for socioeconomic status. The cross-sectional design of the study limited the ability to determine the temporal sequences between tobacco cigarette and EC. Data were collected via self-report.
Kruse 2017 [119]	Cross-sectional survey	68,136 adults with medical comorbidities (8861 with asthma)	EC use was significantly more common in adults current smokers with one or more comorbidities, including asthma, versus those without comorbidities.	All data were self-reported and subjected to recall bias.
Kioi 2018 [120]	On-line based survey	4432 responders with a small fraction suffering from asthma	Ever EC use was not more often among patients with comorbidities.	Selection bias due to being based in an on-line survey. A small fraction from the total number of participants had asthma.
Cooke 2015 [19]	Review	Studies about ECs	ECs could serve as a gateway into tobacco cigarette smoking or illicit drugs (the “gateway effect”).	This review investigated many aspects regarding ECs and was not emphasized in certain ones. This review did not conclude about the effectiveness of EC as a smoking cessation tool.
Han 2020 [68]	Cross-sectional web-based survey	490,171 subjects (44,479 adolescents with physician-diagnosed asthma)	Significantly more subjects had a smoking habit in the asthma group than in the non-asthma group. Both ever and current EC use were significantly associated with alcohol drinking, substance use experience and friends’ smoking. Current EC use was also significantly associated with high caffeine intake in adolescents with asthma.	Case-crossover study: A cause-and-effect relationship could not be established. The self-reported nature introduces the possibility of misclassification in the dataset.
Tran 2020 [75]	Cross-sectional database survey	186,036 adults who responded question about EC use (23,071 with asthma)	Former or current traditional cigarette use were significantly associated with both EC use on some days and every day in adults with asthma.	The self-reported nature introduces the possibility of misclassification in the dataset. Case-crossover study: A cause-and-effect relationship could not be established.
Reddy 2021 [49]	Case series	6 patients (3 with asthma)	6 patients with a median age of 17 years presented with EVALI and half of them had a preexisting diagnosis of asthma. All patients reported tetrahydrocannabinol as well as nicotine EC use.	Case series.
Rodriguez 2020 [107]	Case report	1 patient with a history of childhood asthma	A 23-year-old male, with a history of childhood asthma developed EVALI mimicking COVID-19 disease after smoking marijuana through an EC.	Case report.
Chawla 2020 [108]	Case report	1 patient with a history of mild intermittent asthma	A 15-year-old male with a history of mild intermittent asthma presented with EVALI after using an EC with a new tetrahydrocannabinol (THC) cartridge.	Case report.
Clawson 2020 [66]	Cross-sectional web-based survey	178 college students with asthma	High rates of nicotine and cannabis use among young adults with asthma were found: 37% reporting a lifetime history of using both nicotine and cannabis.	Case-crossover study: A cause-and-effect relationship could not be established.
Alanazi 2021 [78]	Cross-sectional survey	283 youth and young adults from Alabama (151 with asthma)	The frequency of cannabis use in the past 30 days moderated the relationship between asthma and susceptibility to EC use, such that more frequent cannabis use was associated with less susceptibility.	Case-crossover study: A cause-and-effect relationship could not be established.
Thirión-Romero 2019 [16]	Review	Studies about ECs	The rise of EC experimentation among never-smokers, especially children and adolescents, which leads to nicotine addiction and increases the chance of becoming regular smoker, is a major public health issue concern. The prevalence of ever use of ECs has been found to be higher in patients with asthma.	This review investigated many aspects regarding ECs and was not emphasized in certain ones.
Bozier 2020 [32]	Systematic review	11 previously published studies from PubMed	Patients who quit smoking using ECs, continue to vape for long periods after smoking cessation.	The study investigated the effects of EC in general and was not emphasized in a certain group like patients with asthma.
Polosa 2014 [54]	Retrospective observational study	18 asthmatics who switched from tobacco cigarette to EC	18 asthmatic smokers switched from tobacco cigarette to EC with substantial health gains.	Too few patients with asthma for an observational study. Possible selection bias. There is no mention about the original number of asthmatic smokers from which the 18 managed to switch from tobacco to electronic cigarette.
Polosa 2016 [55]	Prospective observational study	18 asthmatics who switched from tobacco cigarette to EC	18 asthmatic smokers switched from tobacco cigarette to EC with substantial health gains. 2 of them (11.1%) later relapsed to tobacco cigarette again.	Too few patients with asthma for an observational study. Possible selection bias. There is no mention about the original number of asthmatic smokers from which the 18 managed to switch from tobacco to electronic cigarette.
Gibson-Young 2020 [77]	Cross-sectional web-based survey	2298 undergraduate college students (446 with asthma)	Approximately 50% of college-age students’ perceived ENDS vapor as less harmful than traditional cigarette smoke. Students with asthma and lower perceived health status reported fewer ever use of ENDS.	Case-crossover study: A cause-and-effect relationship could not be established. Limitation due to the statistical methodology that was used.
Polosa 2016 [98]	Review	Peer-reviewed articles from the PubMed about ECs	Based on existing evidence, EC should be used as a smoking cessation tool when counseling patients with asthma.	Under the heading: “What about Potential Harm?” the authors discussed about ECs’ benefits. Potential harms of ECs were hardly mentioned or were not mentioned at all. Many studies about ECs’ harms were not included in the study.
Farsalinos 2014 [99]	On-line based survey	19,414 (1308 patients with asthma)	0.5% of the participants reported they were non smokers at the time of EC initiation. A reduction in both active smoking percentage and in the number of tobacco cigarette (by approximately 80% in both) was found, after EC initiation.	Selection bias due to being based in an on-line survey.
Goniewicz 2020 [100]	Systematic review and meta-analysis	Six population-based studies (5 cross-sectional and 1 longitudinal–3 reported respiratory outcomes) with sample sizes ranging from 19,475 to 161,529 respondents	Former smokers who transitioned to EC showed ~ 40% lower odds of respiratory outcomes, including asthma, compared to current exclusive smokers.	A small number of mainly cross-sectional studies were included in this systematic review and only three reported respiratory outcomes. The utility of cross-sectional studies for causal inference is limited.
Gugala 2021 [101]	Systematic review and meta-analysis	45 studies (14 randomized experimental, 7 non-randomized experimental, 6 cohort, and 18 cross-sectional) including 1,465,292 participants	EC use resulted in improved outcomes when compared with tobacco cigarette use or dual use of tobacco cigarette and EC.	Some of the studies which included in this meta-analysis were cross-sectional and their utility for causal inference was limited.
Xian 2021 [93]	Systematic review and meta-analysis	11 previously published cross-sectional studies including 1,143,118 participants	When ECs were used in combination with traditional cigarettes, the association odds with asthma was higher than that of users who used traditional cigarettes.	The utility of cross-sectional studies for causal inference is limited.
St Claire 2020 [61]	Review	Previously published studies	The most common form of EC use is the combination with smoking conventional cigarettes. The evidence on the use of EC as potential cessation aid is inconclusive.	The study was oriented more towards tobacco cigarette and not EC.
Franks 2018 [122]	Review	Articles that investigated the effectiveness of EC as a smoking cessation tool in various populations	Additional research is needed for the evaluation of EC for smoking cessation in patients with pulmonary diseases, including asthma.	High-quality studies were lacking to support EC use for smoking cessation. Long-term safety data about EC were also lacking. The study investigated the EC for smoking cessation in various populations and was not oriented in patients with asthma.
Concluding paragraph: The majority of the studies demonstrate the decreased effectiveness of EC in quit smoking by exhibiting two things: (1) Asthmatic dual users smoke more tobacco cigarettes every day than their non-asthmatic counterparts; (2) Asthmatic patients could become addicted to EC easier than non-asthmatics (possible “gateway effect”). On the other hand, the studies which suggest that EC could promote smoking cessation in asthmatic patients, include a small number of participants, or are based on online surveys, a fact that raise serious concerns about a possible selection bias.				

EC = electronic cigarette, EVALI = electronic vapor acute lung injury, CDC = center for disease control, THC = tetrahydrocannabinol, ENDS = electronic nicotine delivery system, COVID-19 = coronavirus disease 2019.

## 4. Discussion

Although several studies showed that ECs may worsen asthma inflammation, there are people who suggest that harm reduction strategy should also apply to asthmatic patients.

As far as we know, most studies concluded that EC acutely worsens lung function of asthmatic patients. Studies that found no difference or even improvement in lung function included a small number of participants and exhibited serious methodological errors [53,54,55]. Airway inflammation was also found to be altered, mainly the Th2 inflammatory pathway, but not limited to that [45,47,50,51].

Most studies indicate the negative effects of vaping on asthma indirectly by the increased likelihood of a vaper being also an asthmatic, with a dose-dependent manner. However, most of the studies were cross-sectional, thus they could not establish a cause-and-effect relationship between EC use and asthma. Nevertheless, they provide excellent epidemiological data to assess trends and note areas where interventions are needed. Studies suggesting that EC improves asthma control presented serious concerns about a possible selection bias, as they were based on online surveys data, or included a small number of asthmatics. Recently published systematic reviews with meta-analyses tally with the above conclusions [93,100,101,102]. More research is needed in order to study the effects of EC on lung function and airway inflammation of asthmatic patients.

As nicotine dependence remains while vaping, most studies reflect the ineffectiveness of EC as a smoking cessation tool by pointing out that patients with asthma could more easily become addicted to EC than non-asthmatics and that asthmatics who are dual users smoke a greater number of tobacco cigarettes per day, while the most alarming finding on this aspect is the ‘gateway effect’. On the contrary, there are also studies suggesting that EC could in fact promote smoking cessation [98,99]. Nonetheless, the study with the most participants supporting that was based on an online survey [99], thus there are serious concerns of a possible selection bias. These conclusions correspond with those of recently published systematic reviews with meta-analyses [93,100,101], and other comprehensive studies [123,124].

## 5. Conclusions

To conclude, this review has consolidated the current knowledge about the effect of EC on asthma by attempting to answer four crucial questions. The studies used in this review had several limitations which are discussed in detail in the tables supporting each question. However, the majority of the studies that used firm laboratory techniques (experimental studies), accurate scientific methodology (interventional studies), and a great number of participants (observational studies) concluded that EC has negative effects on asthma. Studies reporting positive aspects about EC and asthma, presented important limitations, mainly the small number of participants and the possible selection bias. Moreover, many of the authors of these studies presented substantial conflict of interest. It has been shown that a conflict of interest is strongly associated with tobacco industry–favorable results, indicating no harm of ECs [125]. Furthermore, in a blinded assessment, almost all papers without a conflict of interest found potentially harmful effects of ECs, whereas there was a strong association between industry-related conflict of interest and tobacco- and e-cigarette industry favorable results, indicating that ECs are harmless [56]. In any case, as EC is becoming more and more popular, there is an urgent need for long-term prospective studies which will reveal its medium- and long-term effects on the health of asthmatic patients.

## Figures and Tables

**Figure 1 jpm-11-00723-f001:**
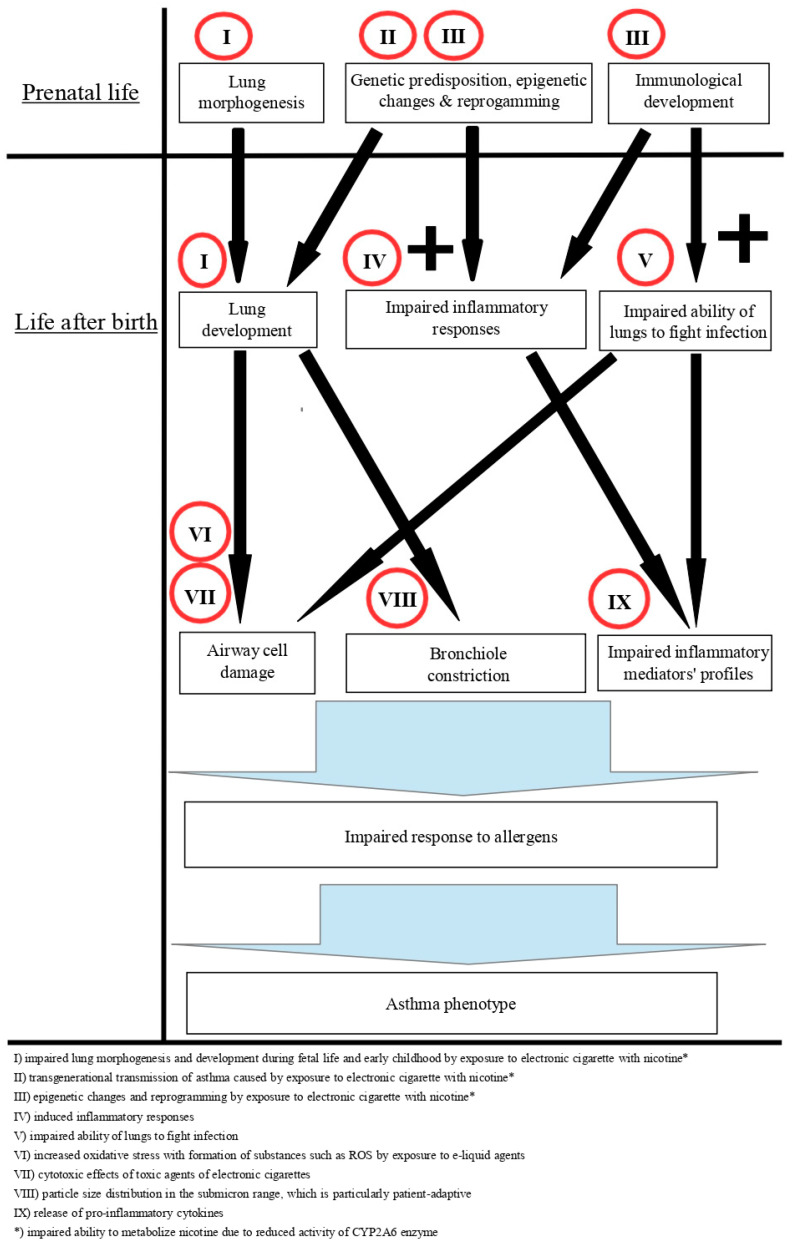
Possible mechanisms whereby electronic cigarette compounds can affect asthma.

## Data Availability

The articles that were used in this review can be found online by using the information that is cited in the reference list.

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
