# Peer review of "Electronic Cigarettes and Asthma: What Do We Know So Far?"

_jpm, 2021, doi:10.3390/jpm11080723_

Round 1

Reviewer 1 Report

An interesting review summarizing a lot of evidence on the association between electronic cigarette use and asthma development and/or control.

  1. I would have liked to see a little more detail on what was the process that led to the elimination of 154 manuscripts identified by the authors’ search strategy (who reviewed them; what was reviewed, the titles, abstracts or other; what irrelevant means etc).
  2. Sentence lines 90-92 is not clear – what does it mean “in the nanometer level”?
  3. It is interesting that the authors include “review manuscripts” in the documents they reviewed here. How were these treated? If a review presented a specific effect of EC, was then the original manuscript describing these effect identified and included in the current manuscript? I would expect there was significant overlap between what was found in review manuscripts and in original research manuscripts. For example, in lines 121-126 the authors present observations from a review (reference 32 in the current manuscript). Was the manuscript that originally present this observation one of the 102 manuscripts reviewed here?
  4. It is not very clear to me what are the mechanisms that the authors propose to affect the development of asthma in response to EC use (presented in Fig 2). What do the “+” signs mean? Does this mean that all changes induced by EC converge in altering response to allergens? How about non-allergic asthma? Also response to allergens is not discussed in the text related to question 1 (lines 79-169).
  5. The studies described in Section 3.2 are the same included in Table 1. However, the way they are organized in the text and the table are different and therefore identifying in the Table the same study you are encountering in the text is not easy. The same is true for Table 2. Both tables seem to be organized based o the year of publication and include together different kind of studies. Maybe another organization scheme would have been easier for the reader.
  6. Lines 241-243 the authors state “Numerous cross-sectional studies with a large number of participants have described 241 the significant association between EC use and even second hand exposure and asthma 242 diagnosis and severity [65-74] …” but the title of this section only talks about asthma control. Maybe the title should reflect the more general evaluation of the role of EC on asthma clinical characteristics that is presented in this section. In this section the authors also present some information on EC and asthma development. Why isn’t this information presented in Section 3.1?
  7. Lines 268-269 “Additionally, asthma was among the most 268 frequently reported side effects associated with EC use, second to headaches” What does it mean that asthma is a side effect of EC?
  8. Lines 291-299 discuss results from meta-analysis of primary studies. The authors do not make clear whether they have already reviewed here the primary studies included in the different meta-analyses they discuss.
  9. Although the manuscript is for the most part easy to follow, I find the writing not very clear in some sections. Paragraphs are not used optimally to focus on specific ideas. In many sections there is no effort to emphasize important conclusions. The document needs some editing for language. For example, the title for section 3.3 (3. What is the effect of using EC on asthma control?) could be more written more clearly. Similar problems are with other sentences across the document. In many cases the meaning of certain sentences is not clear.

Author Response

Response to Reviewer 1 Comments

Dear Reviewer,

We would like to express our thanks for your constructive comments in order to improve our manuscript.

We have revised the manuscript accordingly and have addressed every issue or comment raised, point-by-point as listed below. Any revisions to the manuscript were marked up by using the “Track Changes” function of our MS Word.

Point 1: I would have liked to see a little more detail on what was the process that led to the elimination of 154 manuscripts identified by the authors’ search strategy (who reviewed them; what was reviewed, the titles, abstracts or other; what irrelevant means etc).

Response 1: Thank you for your comment. We have added more details about our search strategy in the “Materials and Methods” section of our revised manuscript as you suggested.

Point 2: Sentence lines 90-92 is not clear – what does it mean “in the nanometer level”?

Response 2: Thank you for your point. The new ENDS devices seem to be particularly user-adaptive, since they produce droplets depending on the vaping pattern of the user. These droplets usually have a diameter below 1μm, which categorizes them “in the nanometer level” (1nm). However, since this phrase -“in the nanometer level”- is confusing, we eliminated it from our revised manuscript.

Point 3: It is interesting that the authors include “review manuscripts” in the documents they reviewed here. How were these treated? If a review presented a specific effect of EC, was then the original manuscript describing these effect identified and included in the current manuscript? I would expect there was significant overlap between what was found in review manuscripts and in original research manuscripts. For example, in lines 121-126 the authors present observations from a review (reference 32 in the current manuscript). Was the manuscript that originally present this observation one of the 102 manuscripts reviewed here?

Response 3: Thank you for your comment. As we described in the “Materials and Methods” section of our manuscript, we used a specific “PubMed search” to identify the studies to use in our review. However, we recognized that this search might have not showed studies that would be relevant to our review. Therefore, we decided to include “review manuscripts” in our study for 2 reasons: 1) In order to miss as less information as possible from other studies that were not showed in our PubMed search (mainly for the 1st question of our review) and 2) In order to cite the findings of other reviews in the subject that we studied and to compare those findings with our own (mainly for the other 3 questions of our review).

Point 4: It is not very clear to me what are the mechanisms that the authors propose to affect the development of asthma in response to EC use (presented in Fig 2). What do the “+” signs mean? Does this mean that all changes induced by EC converge in altering response to allergens? How about non-allergic asthma? Also response to allergens is not discussed in the text related to question 1 (lines 79-169).

Response 4: Thank you for your comment. The “+” signs mean that the effect of EC in the “life after birth” is added to the corresponding effect in the “prenatal life”. More particularly, the “Impaired inflammatory responses” which is an effect of EC after birth, is added to the “Genetic predisposition, epigenetic changes & reprogramming”, which is an effect of EC in the prenatal life. Moreover, the “Impaired ability of lungs to fight infections” which is an effect of EC after birth, is added to the “(impaired) Immunological development”, which is an effect of EC in the prenatal life.

In the same figure we present that EC affects “Lung morphogenesis” in the prenatal life and “Lung development” in life after birth, which along with the “Impaired ability of lungs to fight infections” lead to “Airway cell damage” and “Bronchiole constriction”, which are hallmarks not only of allergic but of non-allergic asthma too. However, we emphasized in the response to allergens since allergic asthma is the phenotype of the majority of asthma cases compared to non-allergic asthma which counts for only a small fraction of asthma cases.

Moreover, it is true that the response to allergens is not discussed directly in the text related to question 1. However, in this text, it is cited in many cases that EC is responsible for impaired immunological responses. More particularly: Lines 105-109: “Heavy EC smoking promotes inflammatory processes ………. complications as asthma [26].” Lines 110-115: “Chronic EC exposure also seems to …….. and in general impaired respiratory innate immune system, all associated with allergies and asthma [27,28].” Lines 115-120: “Respiratory innate immune cell function ……. severity, and/or exacerbations [31].” Lines 123-126: “This increased pro-inflammatory activity suggests ……. cytokine production [32].” Lines 135-138: “Apart from oxidative stress …….. such as asthma [34].” Lines 138-141: “Long non-coding RNAs ……. after exposure to EC vapor [35].” Lines 150-153: “In utero exposure to nicotine-containing EC ……. impairing airway cell function [39].” In all these sentences, we discuss the role of EC in the impaired immunological responses in the lung, which is a hallmark of allergic asthma. Therefore, we believe that the meanings of the text of section 3.1 and of figure 1 complete each other in the discussion about the role of EC in the pathogenesis of asthma.

Point 5: The studies described in Section 3.2 are the same included in Table 1. However, the way they are organized in the text and the table are different and therefore identifying in the Table the same study you are encountering in the text is not easy. The same is true for Table 2. Both tables seem to be organized based o the year of publication and include together different kind of studies. Maybe another organization scheme would have been easier for the reader.

Response 5: Thank you for your point. We re-organized the order in which the studies are appeared in the tables based on the order that they are appeared in the text. We hope that in this way it would be easier for the reader to identify the same study both in the text and in the tables.

Point 6: Lines 241-243 the authors state “Numerous cross-sectional studies with a large number of participants have described 241 the significant association between EC use and even second hand exposure and asthma 242 diagnosis and severity [65-74] …” but the title of this section only talks about asthma control. Maybe the title should reflect the more general evaluation of the role of EC on asthma clinical characteristics that is presented in this section. In this section the authors also present some information on EC and asthma development. Why isn’t this information presented in Section 3.1?

Response 6: Thank you for pointing that out. We used this title because “asthma control” is a key feature of Asthma according to GINA. However, a more general title would better reflect the role of EC on asthma clinical characteristics which are presented in this section, thus we changed the title accordingly.

Additionally, many of the studies that we included in this review, give information about more than one questions of this review. Thus, it is reasonable to exist a certain degree of overlap between the sections of this review. As we mention in the “Materials and Methods” section of our manuscript, some of the studies of this review are included in the answer of more than one questions. However, after carefully reviewing section 3.3 we could not identify any information on EC and asthma development that has not already be mentioned in section 3.1.

Point 7: Lines 268-269 “Additionally, asthma was among the most 268 frequently reported side effects associated with EC use, second to headaches” What does it mean that asthma is a side effect of EC?

Response 7: Thank you for pointing that out. We changed it to: “Additionally, asthma symptoms were among the most frequently reported side effects associated with EC use, second to headaches”.

Point 8: Lines 291-299 discuss results from meta-analysis of primary studies. The authors do not make clear whether they have already reviewed here the primary studies included in the different meta-analyses they discuss.

Response 8: Thank you for pointing that out. In our revised manuscript we clarify that the majority of the studies that were used in those meta-analyses were also reviewed by us in our study. As we mentioned in our response in your 3rd point, we decided to include “review manuscripts” in our study, in order to cite the findings of other reviews in the subject that we studied and to compare those findings with our own.

Point 9: Although the manuscript is for the most part easy to follow, I find the writing not very clear in some sections. Paragraphs are not used optimally to focus on specific ideas. In many sections there is no effort to emphasize important conclusions. The document needs some editing for language. For example, the title for section 3.3 (3. What is the effect of using EC on asthma control?) could be more written more clearly. Similar problems are with other sentences across the document. In many cases the meaning of certain sentences is not clear.

Response 9: Thank you for your comment. We have changed the title for section 3.3 following your recommendation in your 6th point. Moreover, we made some linguistic changes in our revised manuscript, in order to improve it. Finally, we would like to assure you that both our original and our revised manuscripts were last reviewed -before submission- by our co-author Dr. Renata Riha, who is a native English speaker and corrected all word, grammar and syntax mistakes in the manuscripts.

Reviewer 2 Report

In this extensive review, the authors delve into the effects of electronic cigarettes on asthmatic patients. They describe in depth several studies done to date, and provide the main findings and weaknesses of each of them. This is a review that gathers a great amount of information about this field. I have only a few minor comments:

  1. Lines 27-28 (Abstract): “Asthmatic patients should use EC with caution”. In view of the findings and the potential danger that EC has on asthmatics, I believe that this phrase should be deleted as it may lead to confusion about the use of EC.
  2. Lines 57-74 (Materials and Methods): Figure 1 and main text provide the same information (it is repeated). Summarize or eliminate one of them.
  3. Figure 2: Please, change “morpogenesis” by “morphogenesis”.
  4. Although the tables summarize and explain the findings of the studies quite well, I think they are too long and may divert readers' attention. So, I suggest that they be summarized or that at the end of each one of them a concluding paragraph will be placed, summarizing all these findings and that can answer the question proposed in the subheading.

Author Response

Response to Reviewer 2 Comments

Dear Reviewer,

We would like to express our thanks for your constructive comments in order to improve our manuscript.

We have revised the manuscript accordingly and have addressed every issue or comment raised, point-by-point as listed below. Any revisions to the manuscript were marked up by using the “Track Changes” function of our MS Word.

Point 1: Lines 27-28 (Abstract): “Asthmatic patients should use EC with caution”. In view of the findings and the potential danger that EC has on asthmatics, I believe that this phrase should be deleted as it may lead to confusion about the use of EC.

Response 1: Thank you for your comment. We have changed that phrase in our revised manuscript with the phrase “Asthmatic patients should avoid using EC”.

Point 2: Lines 57-74 (Materials and Methods): Figure 1 and main text provide the same information (it is repeated). Summarize or eliminate one of them.

Response 2: Thank you for your point. We have eliminated the Figure 1 from our revised manuscript as you suggested.

Point 3: Figure 2: Please, change “morpogenesis” by “morphogenesis”.

Response 3: Thank you for pointing that out. We have made that correction in our revised figure, which is now “Figure 1”, since we have eliminated the former “Figure 1” according to your suggestion in your previous comment.

Point 4: Although the tables summarize and explain the findings of the studies quite well, I think they are too long and may divert readers' attention. So, I suggest that they be summarized or that at the end of each one of them a concluding paragraph will be placed, summarizing all these findings and that can answer the question proposed in the subheading.

Response 4: Thank you for your comment. We agree with your view about our tables. However, we believe that if we summarize them, many valuable information will be lost for a reader that would like to read our study in detail. Therefore, we added a concluding paragraph to each table, which summarizes all the findings that can answer the question proposed in the subheading, as you suggested.
